



# Changing Ozone Sensitivity in the South Coast Air Basin during the COVID-19 Period

Jason R. Schroeder[1], Chenxia Cai[1], Jin Xu[1], David Ridley[1], Jin Lu[1], Nancy Bui[1], Fang Yan[1,*], Jeremy Avise[1]

[1] California Air Resources Board, 1001 I Street, Sacramento, CA, USA

[*] Now at ICF Consulting, 980 9th Street, Sacramento, CA, USA

*Correspondence to*: Jason R. Schroeder (Jason.schroeder@arb.ca.gov)

**Abstract.** The South Coast Air Basin (SoCAB), which includes the city of Los Angeles and is home to more than 15 million people, frequently experiences ozone ($O_3$) levels that exceed ambient air quality standards. While strict regulation of $O_3$
precursors has dramatically improved air quality over the past fifty years, the region has seen limited improvement in $O_3$ over the past decade despite continued reductions in precursor emissions. One contributing factor to the recent lack of improvement is a gradual transition of the underlying photochemical environment from a VOC-limited regime towards a $NO_x$-limited one. The changes in human activity prompted by COVID-related precautions in Spring and Summer of 2020 exacerbated these already-occuring changes in the $O_3$ precursor environment. Analyses of sector-wide changes in activity indicate that emissions
of $NO_x$ decreased by 15-20% during Spring (April – May) and 5-10% during Summer (June – July) relative to expected emissions for 2020, largely due to changes in mobile source activity. Historical trend analysis from two indicators of $O_3$ sensitivity (the satellite $HCHO/NO_2$ ratio and the $O_3$ weekend/weekday ratio) revealed that Spring of 2020 was the first year on record to be on average $NO_x$-limited, while the "transitional" character of recent Summers became NOx-limited due to COVID-related $NO_x$ reductions in 2020. Model simulations performed with base-case and COVID-adjusted emissions capture
this change to a $NO_x$-limited environment and suggest that COVID-related emissions reductions were responsible for a 0-2 ppb decrease in $O_3$ over the study period. Reaching $NO_x$-limited territory is an important regulatory milestone, and this study suggests that deep reductions in $NO_x$ emissions (in excess of those observed in this study) would be an effective pathway for long-term $O_3$ reductions.

## 1 Introduction

25         The South Coast Air Basin (SoCAB), which is home to the city of Los Angeles and more than 15 million people, has experienced steadily decreasing levels criteria pollutants such as ozone ($O_3$) over the past few decades. However, recent years have been characterized by basin-wide $O_3$ levels that have been flat or even increasing (Aqmd, 2016). Recent literature has suggested that this trend is the result of non-linear changes in the underlying chemistry that creates $O_3$ in the SoCAB (Pollack et al., 2013; Fujita et al., 2013). Therefore, efforts to reduce $O_3$ in the SoCAB need to understand and account for these non-



linearities in the ozone photochemistry and how the photochemical state may change over time as emissions are further reduced (Fujita et al., 2016). In this work, we explore how the photochemical state of the SoCAB changed as a result of the emissions reductions associated with society's response to COVID-19, which resulted in reduced mobile source emissions. This provides a preview of how the photochemical state of the SoCAB may change in the near future, allowing us to better-predict the long-term effectiveness of regulations aimed at reducing $O_3$.

There are no primary emission sources of $O_3$, and instead it is formed through the photochemical interaction between emissions of volatile organic compounds (VOCs) and nitrogen oxides ($NO_x = NO + NO_2$) (Chameides and Walker, 1973; Chameides et al., 1992). However, $O_3$ chemistry is complex and varies non-linearly with respect to precursor concentrations. Therefore, a detailed understanding of the sensitivity of local $O_3$ formation to changes in the precursor environment is essential for drafting effective mitigation strategies. This non-linear response of $O_3$ to concentrations of its precursors results in the presence of two

distinct photochemical regimes, commonly referred to as "$NO_x$-limited" and "VOC-limited" (Chameides et al., 1992; Kleinman et al., 1997; Sillman et al., 1990). Historically, the $O_3$ season in the SoCAB has been characterized as VOC-limited due to an overabundance of $NO_x$, with high $NO_x$ emissions dominated by mobile sources. In such VOC-limited environments, where a significant fraction of the VOCs are from biogenic sources, reduction of $NO_x$ is necessary to achieve long-term ozone reduction but can lead to short-term $O_3$ increases without concurrent action to reduce VOC emissions. Over the past few

decades, reductions in SoCAB $NO_x$ emissions have been accompanied by concurrent reductions in anthropogenic VOC emissions, yielding a general decrease in basin wide $O_3$ (Aqmd, 2016). In recent years, while concurrent VOC and $NO_x$ reductions have continued, VOC reductions have been outpaced by $NO_x$ reductions, offering one explanation for the recent flattening in the $O_3$ trend. Given California's recent initiatives to phase out internal combustion engines in light duty and heavy duty vehicles, which would greatly reduce $NO_x$ emissions, a critical question remains: when will the SoCAB become $NO_x$-

limited and begin to experience immediate benefits in the form of reduced $O_3$ from the reduction in $NO_x$ emissions?
Recent literature has indicated that the SoCAB has been moving away from a VOC-limited environment towards a $NO_x$-limited environment, but discerning the exact nature of the current photochemical state in this "transitional" environment is challenging due to the limitations of individual observation platforms. Surface monitoring networks can be used for spatiotemporal exploration of trace gases and the weekend/weekday (WE/WD) effect, thus providing insight into the

photochemical regime at local scales. The WE/WD effect is a well-studied phenomenon whereby reduced heavy-duty truck activity and emissions on weekends can be employed as a natural experiment to explore the response of $O_3$ to changes in $NO_x$ emissions. In a VOC-limited environment, weekend $O_3$ tends to be higher than weekday $O_3$, and vice-versa for a $NO_x$-limited environment. However, surface monitoring networks are subject to spatial gaps, incomplete temporal coverage, and only represent conditions at the surface ($O_3$ production is integrated throughout the planetary boundary layer). Satellite

measurements, on the other hand, offer greatly improved spatial coverage with daily viewings. Column-integrated measurements of the $HCHO/NO_2$ ratio have been applied as a coarse indicator of $O_3$ sensitivity in the lower troposphere, with very low ratios indicative of VOC-limited regimes, and very high ratios indicating $NO_x$-limited regimes (Martin et al., 2004; Duncan et al., 2010). However, recent studies have shown that this ratio incurs a large degree of uncertainty, and may not be



useful for classifying regimes that are near a "transitional" state (Schroeder et al., 2017). Furthermore, measurements from

polar-orbiting satellites are only collected once per day (typically around midday), meaning that satellite measurements are insufficient to explore diurnal trends in $O_3$ chemistry. Chemical transport models (CTMs) - which include emissions, transport, and chemistry – provide the most robust method of studying $O_3$ chemistry by allowing users to explore changes in simulated $O_3$ in response to changes in precursor emissions. However, CTMs are subject to uncertainties in many parameters (particularly emissions), leading to uncertainty in the simulated dependency of $O_3$ on its chemical precursors. Rather than rely on any one

of these approaches, this study uses a multi-perspective approach whereby all three data sources (satellite data, surface monitors, and a CTM) are integrated into our analysis to paint a cohesive picture of the $O_3$ photochemical regime in the SoCAB during the COVID-19 period.

In the SoCAB, mobile sources are estimated to be responsible for more than three quarters of all $NO_x$ emissions, with heavy-duty trucks comprising the largest sub-sector for mobile source $NO_x$ emissions (CEPAM, 2018). Recent legislature passed by

the State of California aims to eliminate new sales of light duty internal combustion engines by 2035 and heavy-duty internal combustion engines by 2040. Given the drastic $NO_x$ reductions expected by these programs, there is considerable interest in classifying the current photochemical state of the SoCAB and understanding when the region may transition to a $NO_x$-limited environment, thus maximizing the $O_3$ benefit of these policies. California began implementing a 'stay home' policy in March of 2020 in an effort to combat the spread of COVID-19. As a result of this policy, mobile source activity temporarily dropped

in Spring and early Summer of 2020, providing a potential glimpse into how SoCAB $O_3$ chemistry may look in the near future. Recent literature examining the COVID-19 period has highlighted that the SoCAB experienced a 20-40% decrease in ambient $NO_x$ concentrations, no discernable change in ambient VOC concentrations, and inconsistent changes in $O_3$ across the basin (Parker et al., 2020; Goldberg et al., 2020; Barletta et al., 2020; Naeger and Murphy, 2020). This work builds upon these studies by using a suite of indicators to derive process-level understanding and establish causal relationships between

emissions, photochemical state, ambient concentrations, and meteorology during the COVID-19 period in the SoCAB. We show that the additional $NO_x$ reductions associated with COVID-19 were on average sufficient to shift $O_3$ chemistry in the basin into a $NO_x$-limited regime for the first time since observations began. However, changes in chemical regime alone were not enough to reduce ambient $O_3$ concentrations, especially when coupled with warmer-than-usual temperatures observed during the study period.

**2 Methodology**

**2.1 Quantifying Changes in On-road and Off-road Mobile Source Activity**

Although COVID-19 resulted in many behavioral changes among residents of the SoCAB, none had a greater effect on emissions of $O_3$ precursors than reductions in vehicle activity. This study quantifies the changes in vehicle miles traveled (VMT) using vehicle activity monitoring and tracking data from a suite of publicly available sources to quantify changes due

to COVID-related precautions. These data sources include (1) VMT from StreetLight Data, Inc., (Streetlight, 2020) (2) VMT





from Caltrans Performane Measurement System (PeMS) (Caltrans, 2020b), (3) truck counts from Weigh-in-Motion (WIM) stations (Caltrans, 2020a), (4) relative changes of vehicle trips from Geotab, and (5) diesel and gasoline fuel sales from California Department of Tax and Fee Administration (CDTFA).

County-wide data from Streetlight was used to evaluate total VMT trends from March 1, 2020 to the end of July 2020 (Streetlight, 2020). Total VMT is estimated as mean trip length and the total number of trips taken by the full population. To cross-validate Streetlight VMT estimates, VMT from Caltrans' PeMS was computed with data collected in real-time from nearly 40,000 individual detectors spanning the freeway system across all major metropolitan areas of California (Caltrans, 2020b). Though PeMS data is not directly used for emission estimates in this analysis due to its limited vehicle activity coverage (e.g. only highways), it was used to cross-validate Streetlight data. Both datasets showed similar trends of total VMT

reduction during the study period.

Heavy-duty trucks are responsible for nearly one third of all mobile source $NO_x$ emisions in the SoCAB, thus special care must be taken to ensure accurate quantification of VMT from heavy-duty vehicles. In this analysis, daily truck counts with Federal Highway Administration (FHWA) Class 4 and above from Weigh-in-Motion (WIM) stations are used as surrogates to reflect the changes of heavy-duty vehicle activity due to COVID-19 shelter-in-place. Electronic sensors at WIM stations

capture and record truck counts, as well as their axle and gross vehicle weights, when vehicles drive over a measurement site (Caltrans, 2020a). WIM stations with missing data during critical days of observation were excluded from this study, except for the station the 710 freeway, where the missing data were filled based on the average regional contributions (50%-60% during weekdays and 20%-30% during weekends) and observed total region total. Commercial heavy-duty truck trip trends from Geotab were used to cross check WIM data. Despite that Geotab data is limited by telematics data from commercial

fleets they manage, it shows similar pattern to WIM truck counts.

The relative impacts of COVID-19 on off-road mobile source activities were also evaluated from port, railway, and aviation sectors. Port and rail activities were estimated as a relative reduction from prior year using container counts provided by the the port of Los Angeles, port of Long Beach, and freight rail companies. Aviation impacts were estimated as relative daily change from pre-pandemic baseline using Geotab data and counts provided by Los Angeles International Airport. Other off-

road sectors such as agriculture, construction, and other off-road equipment were not evaluated in this study.

## 2.2 Estimating Changes in Mobile Source Emissions

EMFAC2017 (Emfac, 2017) was used as the foundational framework to provide a baseline on-road vehicle emission inventory in 2020 (i.e. estimated emissions for 2020 in the absence of COVID-19). Note that EMFAC2017 uses historical vehicle registration data until 2016, and vehicle activities in 2020 were forecasted. To calculate changes in emissions due to

COVID-19, Vehicle activity, or VMT scalers, relative changes to the baseline in January 2020, were generated based on VMT data as described in Section 2.1. In this study, it was assumed that the scalers of total VMT is representative of light-duty vehicle activity trends, as EMFAC2017 shows that 94% of the total VMT in California is from light-duty vehicles in 2020.



The VMT scalers were then applied to the EMFAC2017 baseline emission inventory in the SoCAB by vehicle class and by emission process. The changes in VMT were used as a surrogates to estimate changes in emissions from processes of running and idling exhaust, evaporative running loss, and brake wear and tire wear. Emissions for the rest of processes are not significantly influenced by vehicle distance traveled, and they are assumed to be the same as the case without COIVID-19 impact. It should be noted that emission rates vary by vehicle speed. With fewer vehicles on road, the average vehicle speed was observed to be higher during the study period. However, the impact of changes in vehicle speed was not included in this analysis due to lack of sufficient data.

**2.3 Surface Monitoring Networks**

In-situ measurements of $O_3$ and $NO_2$ from SoCAB monitoring sites shown in Figure 1 were accessed from CARB's Air Quality and Meteorological Information System (AQMIS2; https://www.arb.ca.gov/aqmis2/aqdselect.php). For $O_3$, data from 2000 through the end of the study period (July 2020) were accessed. Max Daily 8-hour $O_3$ (MDA8) values were calculated for each site for each day. The ratio of WE/WD O3 was calculated at each site using the ratio of period-average Max Daily 8-hour (MDA8) $O_3$ for Sundays versus Wednesdays. This methodology follows previous literature, which noted that Sundays tend to have the lowest VMT, while Wednsedays tend to have the highest (i.e. the difference between "Weekend" and "Weekday" emissions is maximized by using Sundays and Wednesdays as representative days) (Heuss et al., 2003; Yarwood et al., 2003; Wolff et al., 2013).

**2.4 Satellite Data**

To provide a broader spatial context for this study, column-integrated measurements of HCHO and $NO_2$ were obtained from the Ozone Monitoring Instrument (OMI) onboard NASA's Aura satellite. Daily L2 data products were obtained over California and filtered using the data quality flags and recommended QA/QC procedures (Readme, 2019; Lamsal et al., 2021; Readme, 2014). Both satellites provide daily observations at approximately 13:30 local time. For both OMI and TROPOMI, data were obtained from instrument launch through 2020 (i.e. 2005-2020 for OMI, 2018-2020 for TROPOMI). Both instruments provide pixels that vary in size depending on viewing geometry, but are a minimum of 13 x 24 km (OMI) and 3.5 x 5.5 km (TROPOMI) when viewed at nadir. For this work, we calculated the $HCHO/NO_2$ ratio for each observation (i.e. each pixel for each day) for each instrument. Additionally, much of this work utilizes daily spatial averages of satellite products, that is, the average value of all pixels contained within the SoCAB boundary on a given day. However, because $NO_2$ has a lognormal distribution, daily spatial averages of $NO_2$ and $HCHO/NO_2$ can be heavily skewed by the presence of cloudy or partly cloudy scenes. To account for this, data were smoothed by the following process: first, L2 data were temporally averaged to a fixed grid over a 15-day moving window centered on the measurement date (this provides a time-averaged map, 5 x 7 km for TROPOMI, and 13 x 24 for OMI). Then, the mean (for HCHO) or logarithmic mean (for $NO_2$ and $HCHO/NO_2$) of all moving-average grid cells that fell within the SoCAB boundary was calculated.


## 2.5 Model Configuration

Air quality model simulations over California from Feb 23 to July 5, 2020 were conducted using Community Multiscale Air Quality (CMAQ) version 5.2.1 (Appel et al., 2013). The model runs for the first 7 days were considered as spin-up runs and excluded from the data analysis. SAPRC07TC and AERO6 mechanisms were used for gas and particle phase representations, respectively, in the CMAQ model. The modeling domain covers all of California and Nevada as well as part of the Pacific Ocean to the west. The modeling domain is comprised of 321 x 291 horizontal grids with a resolution of 4 x 4 km$^2$. The vertical

grid is represented by 30 vertical layers from the land/ocean surface to 100mb. Default CMAQ initial conditions were used for the model simulation. Chemical boundary conditions were derived from global chemical transport model Goddard Earth Observing System Model version 4 (MOZART-4) simulations conducted at the National Center for Atmospheric Research (NCAR)(Emmons et al., 2010). NCAR discontinued MOZART-4 simulations after January 2018 (https://www.acom.ucar.edu/wrf-chem/mozart.shtml), so 2017 data were mapped to the 2020 calendar, so the global impact

of COVID-19 pandemic is not considered in these model simulations. Weather Research and Forecasting model (WRF) version 3.4 was used to provide meteorology fields as input for the CMAQ simulations (Skamarock et al., 2008). In the WRF model simulation, three nested domains with 36 x 36 km$^2$, 12 x 12 km$^2$ and 4 x 4 km$^2$ horizontal resolutions were employed. Outputs from the inner most 4 x 4 km$^2$ WRF domain were processed by MCIP version 4.3 to drive the CMAQ model.

To investigate the potential impacts of California's response to the COVID-19 pandemic on air quality in SoCAB, two sets of

day specific 2020 emission inventories were prepared for this study. One represents the baseline emission inventory with business as usual and no COVID-19 pandemic related adjustments. The second uses the 2020 baseline emissions as the starting point and then applies COVID-19 related adjustments to the on-road and off-road mobile emissions as described in Sections 2.1 and 2.2. Emission categories include on-road mobile, off-road mobile, area, elevated point, road dust and ocean going vessels. Biogenic emissions were prepared using the Model of Emissions and Gases and Aerosols from Nature v3.0 (MEGAN

v3.0) (Guenther et al., 2006). Due a lack of leaf area index (LAI) data for 2020 at the time of study, 2018 LAI data with 2020 meteorology from WRF were used in MEGAN to estimate biogenic emissions.

In addition to the 2020 simulations, modeling of the 2010 CalNex study and previous regulatory related modeling activities at CARB are presented for 2010, 2012, 2015 and 2017 to study the long term trend of modeled WE/WD compared to observed WE/WD trends (Cai et al., 2019). The simulations for the four previous years utilized year-specific emission inventories and

boundary conditions with CMAQv5.0.2. Other model configurations/settings were consistent with those for the 2020 simulations



# 3 Results

## 3.1 Changes in Precursor Emissions

### 3.1.1 Bottom-up: Impact of VMT on Emissions

Daily total emissions of $NO_x$ and VOC in the SoCAB from baseline and COVID-19 adjusted emission inventories for the modeling period are shown in Figure 2. For $NO_x$, the emissions with and without COVID-19 adjustment both show significant weekday vs. weekend variations with weekend emissions nearly 30-35% less than the weekday emissions. Compared to the baseline emissions, the decrease of $NO_x$ emissions with COVID-19 adjustment started in early March, reached the maximum of -25% in early April, and slowly returned to around -5% by the end of June. The purple line in the top panel of Figure 2

shows the reduction of $NO_x$ emissions due to COVID-19 was generally higher on weekends than weekdays. Total VOC emissions were quite flat from March through the middle of April, when biogenic emissions generally contributed less than 50 tons per day towards total VOC emissions, or approximately 10-15% of the total VOC emissions. Starting from late April, there was a large increase in total VOC emissions which was due to the significant increase of biogenic emissions triggered by warmer temperatures. Biogenic emissions have large day-to-day variability, but averages ~230 tons per day over Summer,

or aproximately one third of the total VOC emissions in the SoCAB. The percentage reduction of total VOC emissions due to COVID19 was much smaller compared to that for $NO_x$ emissions. The maximum decrease of VOC emissions was around -6% in early April. There is no clear weekend vs. weekday variation for VOC emissions.

### 3.1.2 Top- down: Changes in ambient $NO_2$ after adjusting for meteorology

To validate the bottom-up emissions described in Section 3.1.1, we explored methods to provide a top-down estimate of the

change in ambient $NO_2$ resulting from the COVID-19 response. The top-down estimate is complicated by the fact that changes in ambient $NO_2$ during 2020 are a combination of changes in meteorology, chemistry, and emissions. Historically, $NO_2$ concentrations in SoCAB decrease between February and June (See Figure 3); therefore, it is misleading to attribute the decline in $NO_2$ after the onset of pandemic to emissions changes alone. Furthermore, the timing of the seasonal decline in $NO_2$ varies from year to year, so comparison to the same month from previous years is problematic. For example, we compare the average

$NO_2$ concentration between March 20th and June 20th in SoCAB (based on 21 sites with data through 2015 - 2020) for 2020 with the average $NO_2$ from each year between 2015 and 2019. This yields a wide range of -1.5 to -4.4 ppb for $\Delta NO_2$ in 2020, relative to previous years. We improve on this estimate by using the Vehicle Miles Travelled (VMT) data as a predictor of activity change to provide a top-down estimate of the $NO_2$ changes attributable to emissions.

We construct a simple multivariate linear model to predict daily $NO_2$ concentrations based on relative humidity, wind speed,

temperature, day of year, and day since January 1st, 2016 (to capture any long-term trends). The model is trained on hourly $NO_2$ data from 2016 – 2019 and used to predict daily $NO_2$ concentrations in 2020. The model predicts daily $NO_2$ in 2020 well ($r^2 = 0.82$, RMSE = 2.3 ppb); however, the model is typically biased high from March onwards. If we include VMT as a predictor variable (still only training on 2016 - 2019 data) the model more closely fits the observations in 2020 ($r^2 = 0.88$,





RMSE = 1.8 ppb). Figure 4 (a) shows the time series of $NO_2$ from the observations and the model with and without VMT (left
panel). In years prior to 2020, the model without VMT predicts the observed $NO_2$ to within 0.4 ppb and the addition of VMT typically makes only a small difference (Figure 4 (b)). In 2020, the model without VMT overestimates the $NO_2$ by 1.4 ppb relative to the observations. Including VMT provides a large correction to the model, underestimating by -0.4 ppb. We use the bounds to estimate that the $\Delta NO_2$ resulting from emissions changes associated with the pandemic response are -1.4 to -1.8 ppb (or about 19% to 26%; the upper bound is the difference between the two models and assumes that the non-VMT model may
overcompensate).

In parallel with the multivariate model, we use the historical relationship between detrended (7-day rolling average) VMT and $NO_2$ to infer the $\Delta NO_2$ resulting from the $\Delta VMT$ (-17%) during the analysis period. The resulting $\Delta NO_2$ via this method is -1.4 to -2.0 ppb, where the range is determined by the 95% confidence bounds of the Theil-Sen regression of detrended VMT and $NO_2$. The range is almost identical to the result from comparing the multivariate model with and without VMT, above.
This contrasts with the estimate based on historical comparison alone (-1.5 to -4.4 ppb) that suggests a much larger upper bound to the effect of the pandemic response on $NO_2$.

Finally, we compare the top-down estimate of the $\Delta NO_2$ for 2020 with the $\Delta NO_2$ from the 2020 CMAQ emissions inventory described in Section 3.1.1. The CMAQ $\Delta NO_2$ is the difference between two simulations, both use the same meteorology but with 2017 versus 2020 emissions, averaged over the 21 South Coast sites used for the top-down comparison (and the same
March 20th – June 20th time frame). The model produces a $\Delta NO_2$ of -1.1 ppb when updated to 2020 emissions. This is slightly below the top-down estimated range of -1.4 to -2.0 ppb. However, considering that 2020 meteorology was not used this suggests that the 2020 emissions used for the CMAQ modeling are reasonable.

### 3.2 In-Situ and Remote Sensing Observations

In Section 3.1, we quantified the changes in the $O_3$ precursor environment using a bottom-up and a top-down approach. These
two approaches highlight that the abundance of $NO_x$ was diminished in the April-July period of 2020, due to reductions in mobile source emissions. This sections uses satellite data and surface monitoring networks to explore how this drastic change in precursor abundances affected regional $O_3$ chemistry.

### 3.2.1 Satellite HCHO/$NO_2$ as an Indicator of $O_3$ Sensitvity

Over polluted areas, both HCHO and $NO_2$ have vertical distributions that are heavily weighted toward the lower troposphere,
meaning that column-integrated satellite measurements of these gases are fairly representative of near-surface conditions. Many studies have taken advantage of these favorable vertical distributions to invesitigate surface emissions of $NO_x$ and VOCs from space (Duncan et al., 2016; Krotkov et al., 2015; Duncan et al., 2010; Martin et al., 2004; Fishman et al., 2008). Recent literature has shown that the HCHO/$NO_2$ ratio can be a useful indicator of regional $O_3$ sensitivity, although the uncertainty associated with the technique is sufficiently high such that use of the ratio should be reserved for qualitative evaluation of
spatiotemporal trends (Schroeder et al., 2017; Jin et al., 2020). In general, very low ratios are associated with VOC-limited



conditions, and very high ratios are associated with $NO_x$-limited conditions. In Figure 5 (left panel), we show that the OMI $HCHO/NO_2$ ratio in the SoCAB generally increased from 2005 – 2020. Given that previous studies had identified the SoCAB as VOC-limited during the mid 2000's, we can conclude that the increasing $HCHO/NO_2$ ratio in Figure 5 indicates that the SoCAB has been becoming less VOC-limited over time, which is consistent with recent literature (Pollack et al., 2012).

However, this information alone is not enough to conclude whether the region had shifted to a $NO_x$-limited environment by the end of the time series. Figure 5 (right panel) shows the de-trended seasonality of the OMI $HCHO/NO_2$ ratio from 2005 – 2020. In general, lower $HCHO/NO_2$ ratios occurred during Spring compared to Summer, with the increase and plateau corresponding to the seasonality of biogenic emissions in the region (Dreyfus et al., 2002; Misztal et al., 2014). This suggests that, for a given year, Spring (April – May) would appear more VOC-limited than Summer (June – July), and vice-versa.

While Figure 5 demonstrates the qualitative utility of the $HCHO/NO_2$ ratio for exploring interannual and seasonal trends, classification into $NO_x$-limited or VOC-limited regimes can be achieved by coupling this ratio with $O_3$ data from surface monitoring networks. Figure 6 utilizes daily $HCHO/NO_2$ ratios (spatially aggregated over the SoCAB) coupled with daily MDA8 $O_3$ data from the surface monitoring network. These data are binned into five-year increments and separated by season. When presented this way, increasing and decreasing trends within a bin can be used to identify VOC-limited versus $NO_x$-

limited regimes, respectively. In a region that is firmly VOC-limited, $HCHO/NO_2$ ratios are expected to have a positive relationship with MDA8 $O_3$ – that is, days with higher $O_3$ are expected to occur on days with higher $HCHO/NO_2$ ratios (i.e., closest to the "transitional" regime where $O_3$ production is maximized along a ridgeline). In a $NO_x$-limited environment, the opposite would be expected, where high $O_3$ days are expected to occur on days with lower $HCHO/NO_2$ ratios (in a $NO_x$-limited environment, lower $HCHO/NO_2$ ratios would be closer to the "transitional" state where $O_3$ production is maximized along a

ridgeline). Using these expected tendencies as indicators, we can identify seasons and years that were VOC-limited, such as Spring 2005 – 2009, Summer 2005 – 2009, and Spring 2010 – 2014. Spring of 2015 – 2019 and Summer of 2010 – 2014 had no apparent trends (likely indicative of "transitional" states), while Summer 2015 – 2019 showed a slight negative trend, which could be indicative of a weakly $NO_x$-limited photochemical regime. In contrast, both Spring and Summer of 2020 showed strong negative trends, with high $O_3$ days associated with lower $HCHO/NO_2$ ratios. Taken together, this implies that Spring of

2020 was likely the first Spring season to be $NO_x$-limited since records began, while Summer likely moved from a weakly $NO_x$-limited regime in 2015 - 2019 years to a firmly $NO_x$-limited regime in 2020.

### 3.2.2 The $O_3$ Weekend/Weekday Effect

While the satellite-based approach presented in Section 3.2.1 is useful for diagnosing $O_3$ sensitivity at regional scales, there are subtle nuances that cannot be accounted for using the satellite-based approach alone. First, satellite instruments, particularly

OMI, have relatively coarse spatial resolution and have limited utility for diagnosing sub-regional gradients in $O_3$ sensitivity. While exploring $O_3$ chemistry aggregated to the regional scale is certainly useful, additional information about the location of sub-regional gradients in $O_3$ sensitivity can be useful for understanding policy implications when coupled with knowledge of emission sources, population centers, and typical meteorology. Secondly, polar orbiting satellites overpass once per day,





typically around midday (OMI's orbit passes over California around 13:30 local time), making it impossible to diagnose diurnal

trends in O₃ sensitivity from satellite data alone. Ambient O₃ concentrations typically peak in the late afternoon and are the result of chemical production integrated over the course of the day, with midday chemistry appearing more NOₓ-limited in character than other times of day (due to higher actinic flux, higher temperatures, higher biogenic emissions, and reduced NOₓ emissions relative to morning and evening commute times). Applying WE/WD analysis to surface monitoring networks (described in Section 2) enables exploration of sub-regional gradients in O₃ chemistry. Furthermore, because this approach

only employs MDA8 O₃, WE/WD analysis inherently contains information about the outcomes of diurnally-integrated photochemistry. Figure 7 shows the long-term trend in the WE/WD effect in Springtime (left panel) and Summertime (right panel). O₃ monitor data is available prior to 2005, but the time period and binning method shown in Figure 7 were chosen to be consistent with Figure 6. In Figure 7, blue colors (i.e. weekend O₃ higher than weekday O₃) are indicative of locally VOC-limited conditions, whereby reductions in NOₓ on weekends coincided with higher ambient O₃. Orange colors are an indication

of locally NOₓ-limited conditions, whereby reductions in NOₓ on weekends coincided with lower ambient O₃.

In general, the trends shown in Figure 7 agree with those presented in Section 3.2.1. Spring of 2005 – 2019 appear VOC-limited, though the effect was weakening with lighter shades of blue present in Spring 2015 – 2019. The analysis in Section 3.2.1 labeled Spring 2015 – 2019 as "transitional", while analysis of Figure 7 concludes that this period was likely weakly VOC-limited. Summer of 2005 – 2014 was generally VOC-limited with most monitors having WE/WD ratios below 1.

However, by Summer of 2015 – 2019, many monitoring sites had WE/WD ratios above 1, indicative of NOₓ-limited conditions. The spatially-aggregated approach employed in Section 3.2.1 concluded that Summer 2015 – 2019 was weakly NOₓ-limited at the regional scale, however, using the WE/WD ratios in Figure 7, interesting sub-regional gradients are noted. For example, while most sites in Summer 2015 – 2019 had WE/WD ratios below 1, a concentration of monitoring sites near the urban core of LA and the LA/Long Beach corridor had WE/WD ratios above 1. This region is a strong source of NOₓ emissions from

mobile sources, including HDV traffic from the port of Los Angeles and port of Long Beach. This highlights the difficulty in controlling O₃ in VOC-limited environments: urban core areas may lag the broader region in transitioning from VOC-limited to NOₓ-limited, which may have important outcomes with respect to the spatial distribution of O₃ and O₃ exposure.

In Spring of 2020, every monitor in the SoCAB was characterized as having WE/WD ratios below 1. Surprisingly, this includes the urban core of LA and the LA/Long Beach truck corridor. While O₃ chemistry in Springtime is generally more VOC-limited

in nature than in Summertime (i.e. right panel of Figure 5), the deepest reductions in NOₓ emissions due to COVID-19 were observed in Springtime (up to 25% reduction, i.e. Figure 3), implying that COVID-19-related mobile source reductions were of a large enough magnitude to alter the Springtime photochemical environment. This agrees with the satellite-based analysis presented in Figure 6. In Summertime of 2020, all but two of the monitoring sites had WE/WD ratios below 1. This includes the urban core of LA and the LA/Long Beach corridor, which had WE/WD ratios above 1 in every year prior to 2020. While

Summertime NOₓ emissions were not reduced by as much as Springtime NOₓ emissions in 2020 (i.e. Figure 2), the baseline Summertime photochemistry was already weakly NOₓ-limited (i.e. Summer 2015-2019). This implies that the mobile source





reductions attributed to COVID-19 were sufficient to change Summertime photochemistry from "weakly NO$_x$-limited" to "firmly NO$_x$-limited", including the urban core of LA and the LA/Long Beach corridor.

## 3.3 Modelling Results

While analysis of satellite and surface monitor data reveal that COVID-related NO$_x$ reductions were sufficient to push the SoCAB into a NO$_x$-limited regime in both Spring and Summer of 2020, understanding the implications of this photochemical shift on ambient O$_3$ levels requires further consideration. For example, present-day satellites are only capable of measuring HCHO and NO$_2$ once per day, typically around midday. While these satellite data are nonetheless useful, one must consider that MDA8 O$_3$ is the result of diurnally-integrated O$_3$ production, meaning that the current generation of satellites do not

provide a complete picture. Additionally, meteorology makes interpretation of indicators difficult. It is well-documented that downwind areas in the Eastern part of the SoCAB typically experience the highest MDA8 O$_3$ in the basin; therefore, it can be difficult to connect observed WE/WD ratios at receptor sites to photochemical conditions at source regions. Recent papers have analyzed spatiotemporal trends of NO$_2$ in the SoCAB during the COVID period, and all noted that short-term meteorological variability makes it challenging to draw comparisons against recent years (Naeger and Murphy, 2020; Parker

et al., 2020; Goldberg et al., 2020). Parker et al. (2020) concluded that NO$_x$ reductions during the COVID period were not associated with meaningful changes in SoCAB O$_3$ concentrations, however, short-term meteorological variability can also obscure the effects that short-term changes in O$_3$ sensitivity impart on ambient O$_3$ levels. We expand upon these papers by using a compilation of chemical transport model simulations to disentangle the effects of emissions, chemistry, and meteorology on ambient O$_3$ levels during the study period. In Sections 3.3.1 and 3.3.2, we highlight that these model

simulations were able to accurately capture the long-term change in O$_3$ sensitivity over the past decade, and accurately simulated the O$_3$ precursor environment in 2020. With this information in hand, we use two sets of simulations (base-case and COVID-adjusted emissions) to isolate and quantify the change in O$_3$ that can be attributed to COVID precautions.

### 3.3.1 Multiyear Simulations of the WE/WD Effect on O$_3$

CMAQ has been used to simulate California O$_3$ concentrations for 2010, 2012, 2015, 2017 and 2020 (Cai et al., 2019). The

calculated WE/WD ratios of MDA8 O$_3$ from model simulations for these years are compared with the observed ratios at the monitoring sites in SoCAB for April to July. Calculation of WE/WD ratios follows the procedure described in Section 2. The boxplots in Figure 8 show the variation of observed April-July O$_3$ WE/WD ratios among SoCAB monitoring sites from 2000 to 2020. The solid black lines in the box are the mean WE/WD ratios of all the sites for each year. The red dots in Figure 8 are modeled mean O$_3$ WE/WD ratios for the corresponding years. The 2020 modeled data is from the simulation with COVID-19

adjusted emissions. With year-by-year variations, the long term trend of O$_3$ WE/WD ratios shows a general decrease during the past two decades. Compared to the majority of the earlier years, the observed mean ratios after 2014 are much closer to 1.0 with the mean ratios for 2016, 2018 and 2020 all below 1. Model simulated mean O$_3$ WE/WD ratios are very consistent with the observed mean ratios indicating the modeling system captured the change of chemical regime over the years. The difference



between modeled and observed mean ratios for 2020 is larger than the difference for the other years, with the model predicting
a higher WE/WD ratio than observed. This may indicate that, while the model captured the transition to a $NO_x$-limited
photochemical environment, modeled MDA8 $O_3$ may be slightly less sensitive to changes in $NO_x$ than is observed. Detailed
model performance comparing simulated to observed $NO_2$ in 2020 is shown in the following section.

**3.3.2 Model Simulations for $NO_2$ and $O_3$ during the COVID-19 period**

For the COVID-19 period in 2020, we analysed times series of $NO_2$ and $O_3$ from model simulations as well as a comparison
with observations from ground monitoring sites. The difference between the two model simulations with and without COVID-
19 adjustment reflects the impact of COVID-19 emission reductions. The left top panel of Figure 9 shows the daily average
$NO_2$ concentrations from observations and the model simulations with COVID-19 adjusted emissions. The right top panel of
Figure 9 shows the $NO_2$ difference between the two model simulations (COVID-19 adjusted – baseline). Since observed $NO_2$
at a monitoring site can be heavily impacted by the local emissions but modeled data are diluted concentrations in 4 x 4 $km^2$
grids, for the $NO_2$ model comparison with observation we considered 18 non-near road monitoring sites. As shown in Figure
9, the model captures the day-to-day variations of $NO_2$ reasonably well. The normalized mean biases of $NO_2$ for March, April,
May and June are -24%, -8%, -4% and 1% respectively. The larger model bias in March is due to the significant model under
estimations during the second and third weeks of March when there was rain. The difference between simulated $NO_2$ using
COVID-19 adjusted emission inventory and baseline emission shows that $NO_2$ concentration was reduced by up to 2.6 ppb
due to the COVID-19 emission reduction where the maximum reduction occurred on March 31. During April and early May,
the $NO_2$ reduction was generally between 1 to 2 ppb. After May 10, the reduction of $NO_2$ continued to be smaller and becomes
nearly diminished by the end of June.

Observed and model simulated MDA8 $O_3$ concentrations are shown in the left (bottom) of Figure 9. Larger discrepancies
between model and observation can be seen for the second and third weeks of March as well as the second week of April when
$O_3$ concentrations are generally low. Those two time periods were associated with two rain events, above average cloud
coverage and relative humidity as well as below average temperature. From mid-April to the end of June when $O_3$
concentrations were relatively higher, modeled $O_3$ concentrations are in very nice agreement with the observations. Significant
enhancement of $O_3$ was observed during late April and early May. This $O_3$ enhancement was successfully captured by the
model except that the peak $O_3$ concentrations on May 6 and 7 were under predicted in the model consistent with the under
prediction of $NO_2$ during these days.  During the late April to early May, high pressure ridges were the dominant weather
patterns over SoCAB. The region had weak offshore winds, low planetary boundary layer height and extremely high
temperatures, which favor the production and accumulation of $O_3$. The highest average daily maximum temperature from all
the monitoring sites reached 33.5 °C on May 6 in both the observations and model simulation. The $O_3$ difference between
model simulations with COVID-19 adjusted emissions and baseline emissions is illustrated in the right bottom panel of Figure
9. The COVID-19 emission reduction caused the $O_3$ concentration to increase by up to 1.2 ppb from March to mid-April and
mostly decrease by up to 2 ppb from late April to early July. On May 7 when the highest $O_3$ was observed, the $O_3$ concentrations





were reduced by about 1.7 ppb due to the COVID-19 emission reductions. The change in $O_3$ difference between the two model simulations over time, especially the shift from positive to negative, indicates the transition from VOC-limited chemical regime to more $NO_x$-limited chemical regime.

## 4 Discussion

Using a multi-perspective approach involving satellites, surface monitors, and modeling, we show that the SoCAB was on average $NO_x$-limited during the COVID-19 period in April – July of 2020. While satellite data and the weekend/weekday effect suggest that Summertime in recent years may have been slightly $NO_x$-limited even before COVID-related mobile source reductions, Spring of 2020 was the first Spring on record to display $NO_x$-limited characteristics. This outcome was achieved by relatively large emissions reductions in Springtime (~20% reduction in SoCAB $NO_x$ emissions), which was sufficient to offset the typical climatology of $O_3$ sensitivity in the region. In Summertime, when $O_3$ sensitivity is naturally more $NO_x$-limited than in Spring (a combination of biogenic emissions, warmer temperatures, and higher actinic flux), a 5% reduction in SoCAB $NO_x$ emissions due to COVID-19 acted to push the region further into $NO_x$-limted territory. In both Spring and Summer of 2020, reductions in mobile source emissions due to COVID-related precautions were the largest contributor to regional $NO_x$ emissions reductions. Thus, the natural experiment offered by data collected during the COVID-19 period of 2020 highlights that reductions in mobile source emissions alone could be a feasible pathway for shifting the SoCAB into a $NO_x$-limited $O_3$ production regime.

This work builds on recent studies that focused on the impacts of COVID-related precautions on $O_3$ and its precursors in Southern California. Naeger et al. (2020) compared TROPOMI $NO_2$ levels in Spring of 2020 to levels observed in Spring of 2019. While the authors excluded wet periods from their analysis, the 40% reduction that they report in Los Angeles is higher than similar studies that attempted to better-account for meteorology, such as Goldberg et al. (2020), who reported a 32% reduction in TROPOMI $NO_2$ for Los Angeles. In Section 3, our bottom-up approach using measurements of vehicle activity yielded an estimated $NO_x$ emissions reduction of 25% during the deepest point of the shutdown, with typical reductions of ~15-20% during Springtime and ~5% during Summertime. While our bottom-up estimate of the reduction in $NO_x$ emissions due to COVID-19 is lower than those reported in Naeger et al. (2020) and Goldberg et al (2020), there are important considerations that must be made to address this discrepancy. First, both Naeger et al. (2020) and Goldberg et al. (2020) focused on city-scale observations, which included the city of Los Angeles rather than the broader SoCAB region focused on in this study. Naeger et al. (2020) showed that $NO_2$ reductions were strongest in the urban core, therefore the numbers reported in Naeger (2020) and Goldberg (2020) would likely decrease if expanded to the broader SoCAB region. Secondly, ambient $NO_2$ levels are a function of both emissions and removal. As shown in this work, reduced $NO_x$ emissions produced a fundamental shift in the underlying photochemistry of the region, which could lead to a decrease in the $NO_x$ lifetime due to enhanced photochemical cycling. Therefore, a given decrease in the $NO_x$ emission rate could be concurrent with an increase in the $NO_x$ removal rate, leading to a larger observed decrease in ambient $NO_2$ than can be explained by emissions alone.


Recent work by Parker et al. (2020) also analyzed $O_3$ and its precursors in the SoCAB during the COVID-19 'stay at home

order' by integrating satellite measurements and surface measurements. The authors concluded that the $NO_x$ reductions observed during that period in the SoCAB were not sufficient to reduce $O_3$ levels across the basin, and instead advocate for VOC controls concurrent with $NO_x$ reductions as a pathway for controlling $O_3$. However, Parker et al. (2020) focused on the outcome (i.e., ambient $O_3$ concentrations) and used that to infer about the underlying process (i.e., $O_3$ chemical regime). Our work expands upon the work in Parker at el. (2020) by including the satellite $HCHO/NO_2$ ratio as an indicator of regional $O_3$

sensitivity and by performing model simulations with base-case versus COVID-adjusted emissions. While our conclusions generally agree with Parker et al. (2020), our finding that on average the SoCAB transitioned into a $NO_x$-limited regime in both Spring and Summer of 2020 cannot be understated. While reaching $NO_x$-limited territory is certainly not the same as reaching regional $O_3$ attainment, it is nonetheless an important milestone from a regulatory perspective. The fact that $O_3$ levels in April – July of 2020 were not particularly different from recent years does indeed suggest that $NO_x$ reductions similar to

those observed in 2020 would not be sufficient for meaningful $O_3$ improvements to be realized. However, our modeling experiment suggests that COVID-related $NO_x$ reductions resulted in $O_3$ levels that were 0-2 ppb lower than they would have been in April – July in the absence of COVID-related precautions. The fact that the SoCAB shifted to a $NO_x$-limited regime and experienced a reduction in simulated $O_3$ (per our modeling study) emphasizes that drastic reductions in $NO_x$ emissions (more than the reductions observed in 2020) will be effective in reducing ambient $O_3$ (though response on any given day or

site may differ from the basin and/or seasonal average conditions). This finding is well-aligned with recent state legislation, such as the Heavy-Duty Omnibus Regulation (https://ww2.arb.ca.gov/rulemaking/2020/hdomnibuslowNOx) and the Governor's Exeutive Order mandating that all new light-duty vehicle sales be zero emission by 2035, with heavy-duty sales to follow by 2045 (https://www.gov.ca.gov/wp-content/uploads/2020/09/9.23.20-EO-N-79-20-Climate.pdf). While Parker et al. (2020) is correct that concurrent reductions in VOC emissions will also be beneficial for controlling $O_3$, a large portion of

SoCAB VOC emissions during $O_3$ season are biogenic in nature (approximately one third or more by mass), implying that significant reductions in ambient $O_3$ can only be achieved with drastic reductions in $NO_x$ emissions.

Achieving attainment for $O_3$ air quality standards will be further complicated by the enhanced frequency of drought and heat waves that California is expected to experience due to a changing climate (Swain et al., 2014). While short-term heat waves result in enhanced photochemical activity and are typically associated with the highest $O_3$ values of a season (for example, the

heat wave highlighted in Section 3.3.2), multi-year droughts are believed to inhibit biogenic VOC emissions, which could lead to reductions in $O_3$ production (Demetillo et al., 2019). While this study shows that statewide efforts to drastically reduce mobile source $NO_x$ emissions will be effective for long-term reductions in ambient $O_3$ in the SoCAB, the effectiveness of such policies may be partially obscured by short-term meteorological variability and long-term climate change. Disentangling these climatic effects from the effects of regulations will be a challenge for scientists and policymakers in the next decades. Follow-

up studies will likely require novel fusions of observation systems including climate models, ultra-high-resolution chemical transport models, geostationary satellites, and dense monitoring networks.



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





**Figure 1. Maps of SoCAB surface O₃ and NO₂ monitors (top and bottom, respectively). The outline of the SoCAB boundary is shown**
**as a white polygon.**

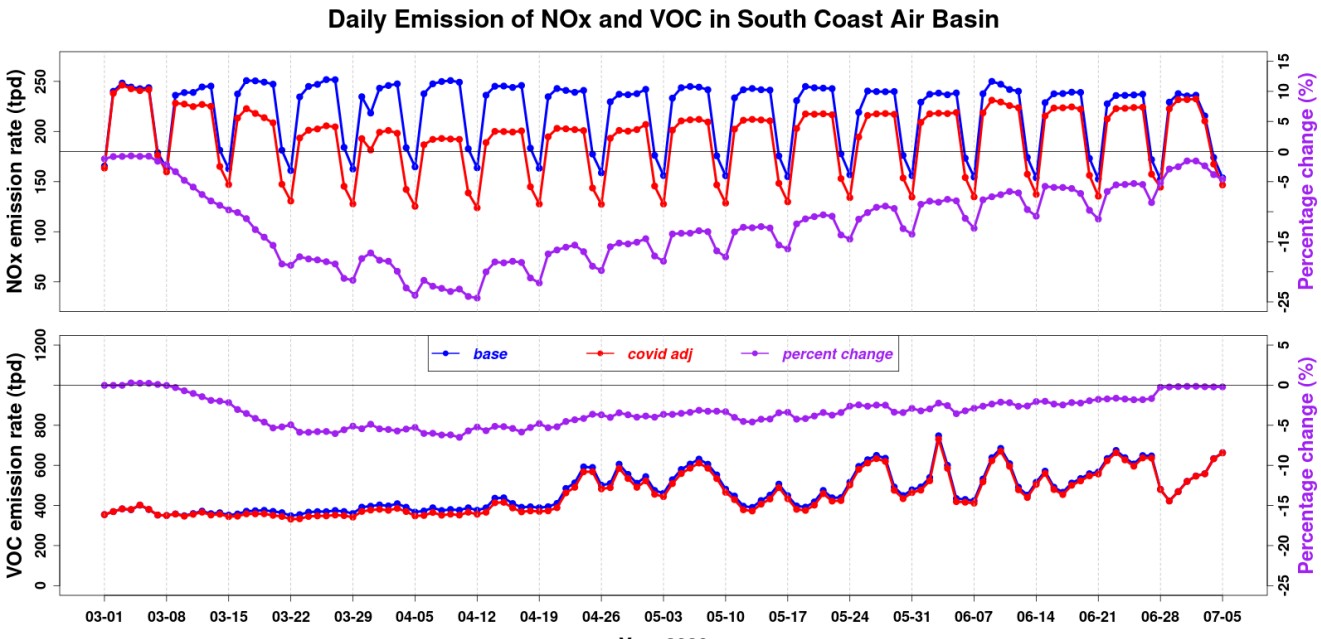

**Figure 2. Daily total emissions of NOx and VOC in South Coast Air Basin. Blue lines are for baseline emissions and red lines are for COVID19 adjusted emissions. Purple lines are for the percentage changes of NO$_x$ and VOC due to COVID19 adjustment: (COV19 adjusted – baseline)/baseline. Vertical grey dash lines correspond to Sundays.**


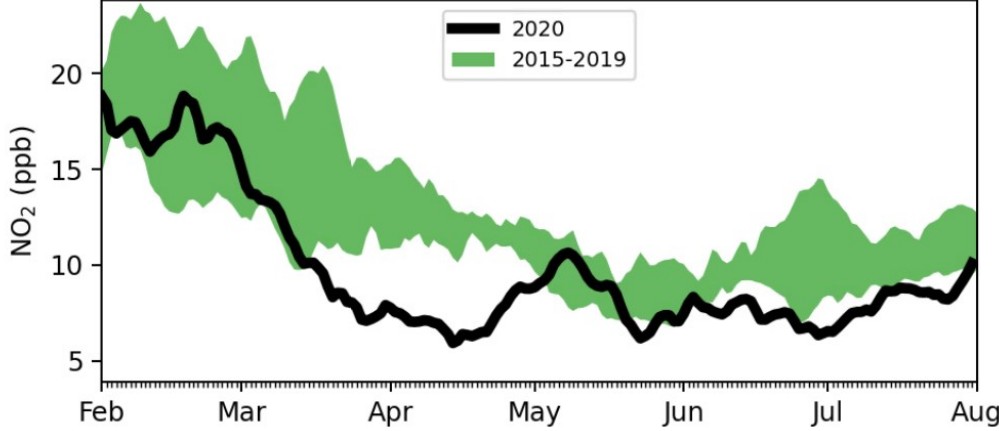

**Figure 3. SoCAB pollutant concentrations for 2020 (black line) and for 2015 – 2019 (green shading). Data is averaged over 21 sites that contain data throughout the time period.**



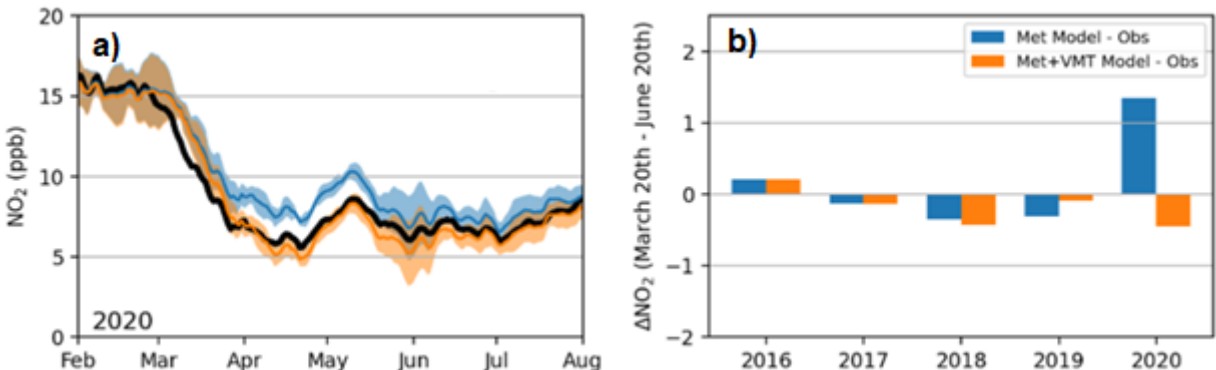

**Figure 4. (a) Observed NO₂ (black) and predictions of daily NO₂ based on non-VMT variables (blue line). Uncertainty estimates are derived from the model-observation difference for the years on which the non-VMT model is trained (narrow envelope means the model performed well on the training data for that time of year). The model with VMT is shown (orange). Data has 21-day smoothing applied. (b) The average difference between the models and the observations for March 20th to June 20th for each year (models are trained on 2016-2019 data).**

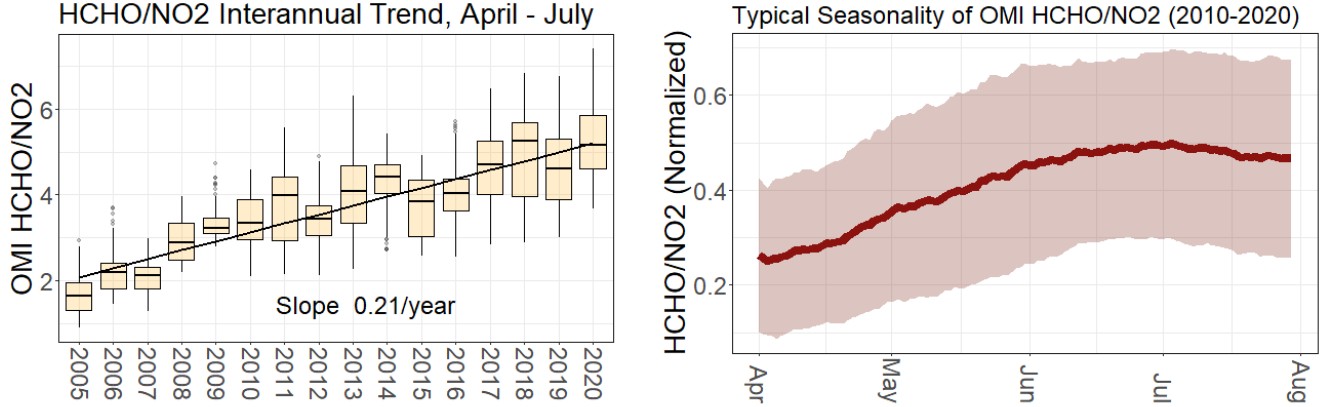

**Figure 5. (Left) Interannual trend of OMI HCHO/NO₂ averaged over the SoCAB. Data are filtered to include the April – July period of each year. A linear fit is applied to the data. (Right) Typical seasonality of OMI HCHO/NO₂ averaged over the SoCAB. Data have been normalized to the range for each year.**



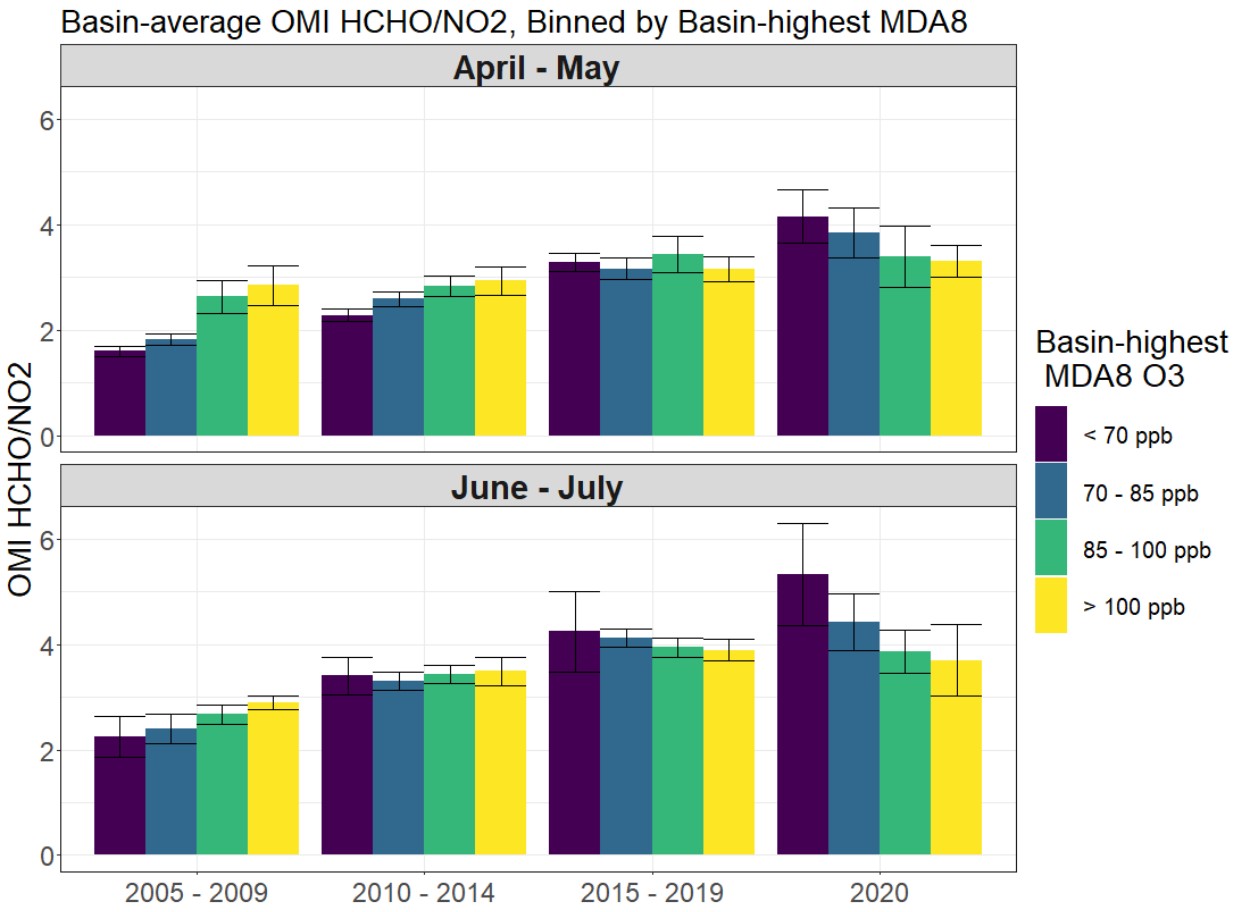

**Figure 6. Time series of basin-average OMI HCHO/NO₂ ratios. Data are binned every five years, and colored by the basin-highest MDA8 observed on each day. Each bar represents the mean ± one standard deviation. The top panel shows data from Spring (April – May), and the bottom panel shows data from Summer (June – July).**





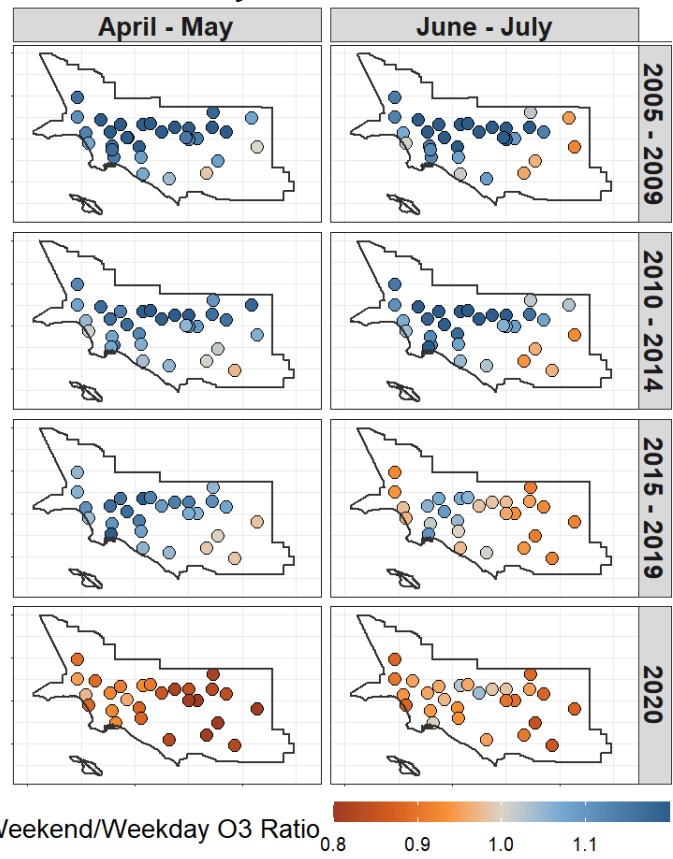


Figure 7. Time series of WE/WD $O_3$ ratios at SoCAB monitoring sites. Rows represent averages over five-year bins, and columns represent temporal bins for Spring (April – May) and Summer (June – July).



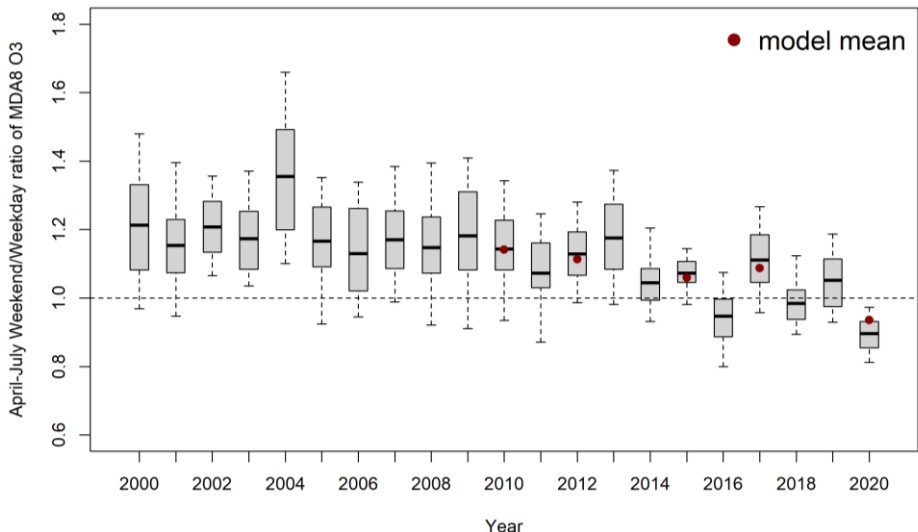

**Figure 8. Boxplot: observed April-July WE/WD ratios of MDA8 O₃ at South Coast monitoring sites. The solid lines in each box are**
**the mean ratios of all the sites. Red dot: modeled mean ratios.**

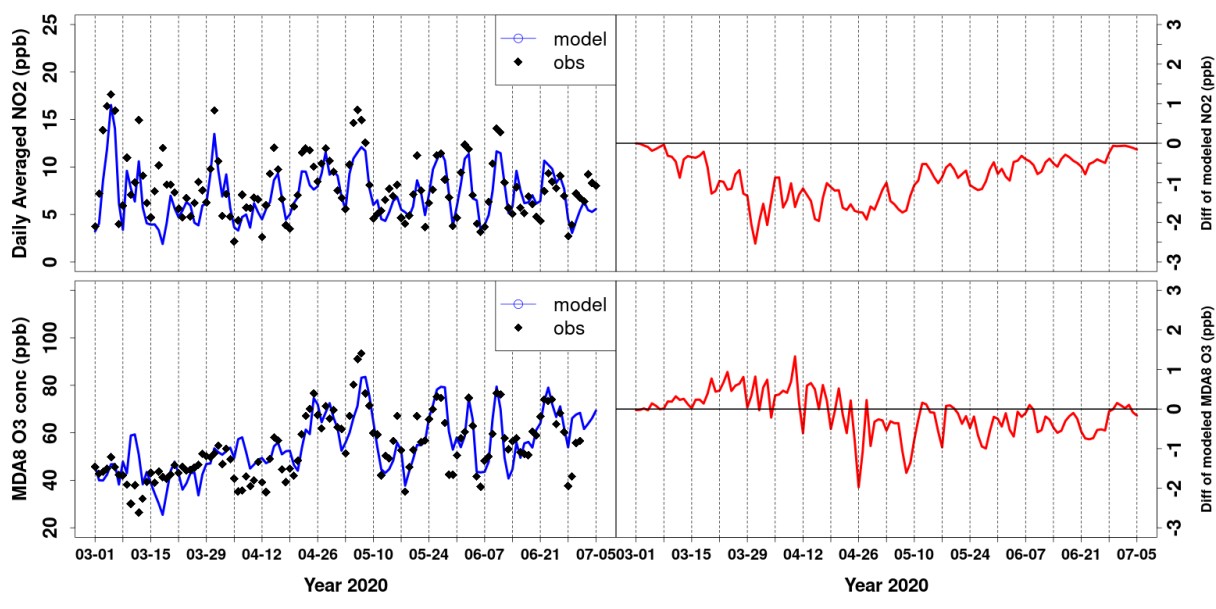

**Figure 9. Left top: time series plots of daily averaged NO₂ concentrations in South Coast Air Basin from model and observation.**
**Right top panel: time series plots of daily averaged NO₂ difference between the two sets of model simulations (model with COVID-**
**19 adjusted emission inventory – model with baseline emission inventory). Left bottom panel: time series plots of daily maximum**
**averaged 8-hour ozone concentrations in South Coast Air Basin from model and observation. Right bottom panel: time series plots**





of MDA8 O$_3$ difference between the two sets of model simulations (model with covid19 adjust emission inventory – model with baseline emission inventory).