# Peer review of "Changing Ozone Sensitivity in the South Coast Air Basin during the COVID-19 Period"

_Atmospheric Chemistry and Physics, 2022_

## Referee Comment (RC2)

Review of " Changing Ozone Sensitivity in the South Coast Air Basin during the COVID-19 Period"

ACP

Current Due Date:  04 May 2022

*General Comments:*

Overall this is an interesting, multi-faceted, and comprehensive paper that is well written and adds to the ever-expanding body of literature on the impacts of the COVID-19 pandemic and lockdown on ozone pollution, in this case for the SoCAB region.  While I think it does provide some important results that would be suitable for publication in ACP, I have a number of general and specific comments related to methods, analysis, and discussion/conclusions that need to be considered below.  My formal recommendation is to reconsider the paper after major revisions.

First, I have some issues with using a 10-year old version of WRF model, but this can be rectified by providing an associated meteorological evaluation to verify usage to drive the CMAQ model.  Also, it wasn't entirely clear if 2020 meteorology was actually used in the WRF simulations used to drive CMAQ.

Second, I feel there is a lack of a necessary supporting document, which should contain a number of supporting analysis to the main results shown.  For example, while there is discussion on the development of COVID-19 transportation-related activity data in the paper (Section 2), I feel these analyses should be shown graphically or in tabular form in a supporting information document.  These activity changes are critical to development of COVID-19 related emissions and ozone precursor changes, and eventual ozone formation modeling results.

Third, there appears to be some conflicts with a recent paper on COVID-19 impacts on ozone sensitivity and concentration changes also in the SoCAB region (Parker et al., 2022), which need to be at least discussed and/or rectified as to the major differences.   Particularly, two of Parker et al. conclusions are the following:

- "Although meteorology played the major role in the increases in ozone between 2019 and 2020, the reduction in NOx emissions due to the response of the COVID pandemic also **caused ozone increases** in Los Angeles County and into western San Bernardino County, with more widespread ozone decreases further east."

- "Ozone formation in parts of the **SoCAB is still VOC-sensitive**, and the locations where NOx reductions cause ozone increases occur in areas with some of the highest population density in the SoCAB".

Which conflict somewhat with the conclusions of this present paper, such as:

- "Model simulations performed with base-case and COVID-adjusted emissions capture this change to a NOx-limited environment and suggest that COVID-related emissions reductions were responsible for a **0-2 ppb decrease in O3** over the study period."

- "Historical trend analysis from two indicators of O3 sensitivity (the satellite HCHO/NO2 ratio and the O3 weekend/weekday ratio) revealed that Spring of 2020 was the first year on record to be **on average NOx-limited**, while the "transitional" character of recent Summers became NOx-limited due to COVID-related NOx reductions in 2020."

While I tend to agree more with the present study findings because they explored the ozone formation/sensitivity using a combination of satellite data, surface monitors, and models, some discussion and comparison on the different conclusions reached in these two studies are necessary here, likely in the "Discussion" section.   This also could be rectified by expanding the modeling analysis to show COVID19-Baseline results spatially across the SoCAB.    There are also some similarities between studies, such as both finding that warmer than average temperatures in the SoCAB played a major role in ozone increases during the COVID-19 lockdown periods of spring/summer 2020.  Please review carefully and provide additional analysis and discussion.

Parker, L.K.; Johnson, J.; Grant, J.; Vennam, P.; Parikh, R.; Chien, C.-J.; Morris, R. Ozone Trends and the Ability of Models to Reproduce the 2020 Ozone Concentrations in the South Coast Air Basin in Southern California under the COVID-19 Restrictions. Atmosphere 2022, 13, 528. https://doi.org/10.3390/atmos13040528.

*Specific Comments:*

1. Lines 51-53:  Please provide a list of citations/references to support these arguments.

2. Lines 81-83:  A new study by Parker et al. (2022) should also be listed here (see General Comments above):

Parker, L.K.; Johnson, J.; Grant, J.; Vennam, P.; Parikh, R.; Chien, C.-J.; Morris, R. Ozone Trends and the Ability of Models to Reproduce the 2020 Ozone Concentrations in the South Coast Air Basin in Southern California under the COVID-19 Restrictions. Atmosphere 2022, 13, 528. https://doi.org/10.3390/atmos13040528.

Furthermore, while not focused on the SoCAB, Campbell et al. (2021) found high spatiotemporal variability in ozone changes (including southern California) during the COVID-19 periods of Spring/Summer 2020 in the U.S.  This may be included as well.

Campbell, P. C., D. Tong, Y. Tang, B. Baker,  P. Lee, R. Saylor, A. Stein, S. Ma, and L. Lamsal (2021).  Impacts of the COVID-19 Economic Slowdown on Ozone Pollution in the U.S. Atmospheric Environment, https://doi.org/10.1016/j.atmosenv.2021.118713.

3.  Lines 104-105:  Should at least these independent VMT dataset calculations be shown as a supporting figure for the reader to get an understanding of how the activity trends change during COVID-19 lockdowns?

4.  Lines 114-115:  Similar to Comment 3, I think the heavy-duty truck trip trend comparisons of WIM vs. Geotab should be shown, at least as supporting figures.

5.  Lines 116-119:  Again, this activity data would be valuable information to show as supporting tables.  Please revise.

6.  Lines 126-127:  If EMFAC2017 forecasts the emissions for 2020, how can it determine that "94% of the total VMT in California is from light-duty vehicles in 2020".  Wouldn't actual data (not projections) be needed to determine this percentage *in* 2020?

7.  Line 164:  There needs to be at least a supporting information figure showing the domain configuration/coverage.

8.  Lines 165-166:  More information is needed on what "Default CMAQ initial conditions" actually represents.  Please revise and expand briefly on this.

9.  Lines 170-171:  I am concerned that a version of WRFv3.4, which is now 10 years old since release date, is used as the driving meteorology.  While I understand the difficulty in always trying to stay state-of-the-science with model versions, I think there should be some effort to at least stay up-to-date with major version changes (i.e. WRVv4). Particularly at high resolution within complex terrain of the west for example, the use of default hybrid terrain-following vertical coordinate in WRFv4 can make an impact on results.

    With that in mind, I would accept the following as a response to this issue:  Include a meteorological evaluation of the WRFv3.4 results, and thus verify and discuss in the paper that use of this WRFv3.4 configuration is acceptable to drive CMAQ here.

10. Lines 171-173:  The three nested WRF model domains should be shown in a figure, and the major WRF model physics selected should be included somewhere in the text or associated table.

    Also, this pertains to a comment below, but was a WRF simulation conducted for the year 2020? What were used for the meteorological initial and boundary conditions, and what year?

    Please be explicit with all this information, and if 2020 meteorology was not used in WRF, then why and what are the potential impacts?

11. Line 173:  Need to define MCIP acronym and further provide a citation/reference.

12. Lines 209-213: Why is applying ground-based NO2 observations and the bottom-up VMT data (from Section 3.1.1) implied here as a "top-down" estimate of NO2 changes? In my opinion, this also leads to a misleading title of this section. Please explain and revise.

13. Lines 215-217: I have some issues with this model being trained on NO2 surface observations, which are known to have systematic issues with interference from other species (e.g., NOy; see Dickerson et al. 2019). I think at least some discussion is needed on this and how it may (or may not in this case) limit the interpretation of the linear model results and upper/lower bounds used to estimate the ΔNO2 resulting from emissions changes due to pandemic response.

    Dickerson, R.R.; Anderson, D.C.; Ren, X. (2019). On the use of data from commercial NOx analyzers for air pollution studies. Atmos. Environ., 215. https://doi.org/10.1016/j.atmosenv.2019.116873.

14. Lines 236-237: This is confusing, as it seems that this is stating that 2020 meteorology wasn't used in the WRF-CMAQ simulations. However, Section 2.5 states that 2020 WRF meteorology was indeed used,and thus seems to conflict with this statement.

    This also affects the conclusion that if 2020 met wasn't used, it suggests the derived 2020 CMAQ emissions were reasonable. If indeed 2020 meteorology wasn't used in the WRF-CMAQ simulations, then this brings up a larger issue as to why this was the case, and why it is not clearly explained in Section 2.5 (See previous comment # 10 above). Please clarify and revise.

15. Lines 260-262: To avoid the issue of biogenic seasonality impacts on the HCHO/NO$_2$, why not also use the more robust O$_3$/NO$_y$ photochemical indicator? I believe while more sparse, there are NOy observations available in the SoCAB.

16. Lines 293-300: There appears to be conflicts in the WE/WD ratio and discussion. On lines 293-294, it is stated "In Figure 7, blue colors (i.e. weekend O3 higher than weekday O3) are indicative of locally VOC-limited conditions, whereby reductions in NOx on weekends coincided with higher ambient O3." While on lines 299- 300 it states "Summer of 2005 – 2014 was generally VOC-limited with most monitors having WE/WD ratios below 1. However, by Summer of 2015 – 2019, many monitoring sites had WE/WD ratios above 1, indicative of NOx-limited conditions." From what I understand, WE/WD > 1 → VOC-limited and WE/WD< 1 → NOx-limited.

17. Lines 332-334: As noted above, an additional evaluation of the simulated meteorology is needed to be confident of the use of WRF modeling to disentangle effects of emissions, chemistry, *and meteorology*.

18. Lines 350-351: Or, this also could be due to high uncertainty in the emissions used for the 2020 COVID-19 period.

19. Lines 362-363: Again, a meteorological evaluation is needed to verify this argument, and it is not clear if the author is referring to an overprediction of WRF clouds/precipitation which may lead to the larger model NO2 underpredictions in second and third weeks of March. The argument "...when there was rain." is too vague and does not provide enough information on the impact/evaluation of simulated meteorology related to the agreement of modeled NO2 and observations.

20. Lines 370-371: Same as in #19 above. How did the simulated WRF meteorology capture these events, and how does this relate to the NO2 and ozone biases?

21. Lines 377-378: OK, there is some mention of WRF met evaluation here, but nothing is shown in the paper, and thus it needs to be expanded upon, e.g., provided as similar diurnal time series plots.

22. Lines 380-384: While the time series analysis averaged over the SoCAB is nice, I think it is limited and does not show the spatial differences between COVID-Baseline across the SoCAB from the model. As shown in Parker et al. (2022), their work shows a distinct gradient in modeled MDA8 ozone due to reduced COVID-19 emissions (see Figure 10 of their work, reproduced below), which indicates for June 01 - July 31 the reduction in NOX emissions caused ozone increases in Los Angeles County and into western San Bernardino County, with more widespread ozone decreases further to the east in the SoCAB. From this standpoint, as mentioned above in the general comments, there seems to be a conflict between results; however, it would be easier to rectify this if a spatial difference plots across the SoCAB are included in the current paper.

[Figure]

**Figure 10.** June 1 – July 31 impacts of the COVID-19 emissions reductions on ozone, estimated by CMAQ: (a) Period average, period maximum, and period minimum MDA8 ozone spatial maps for 2020 COVID and 2020 COVID minus 2020 BAU; (b) Close up of period-average 2020 COVID minus 2020 BAU for SoCAB.

Ozone Trends and the Ability of Models to Reproduce the 2020 Ozone Concentrations in the South Coast Air Basin in Southern California under the COVID-19 Restrictions

Lines 414 - 446:  It is here in this discussion that the new results of Parker et al. (2022) commented on above can, and should be included compared to the present results.

Technical corrections:

1.  Line 112:  Seems like a typo:  "for the station the 710 freeway,".
2.  Line 131:  Typo:  "COIVID-19" should be "COVID-19".
3.  Line 139:  Typo:  "O3" should be "$O_3$"
4.  Line 141:  Typo:  "Wednsedays" should be "Wednesdays"
5.  Line 180:  Typo:  "Due a lack of..." should be "Due to a lack of.."

---

## Author Response (AR1)

Reply to RC2:

Thank you for taking the time to review our draft manuscript. The authors appreciate the reviewer's suggestions and will address each of them.

The reviewer notes that the version of WRF indicated in the draft is ten years old. The authors thank the reviewer for catching this detail - we made a mistake when preparing the draft, and the version of WRF that was used is WRFv4.2.1 (far less old). We will make this correction in the text. The authors also appreciate the reviewer's suggestion for adding a meteorological evaluation, and we have prepared a Supplemental Information document that includes a table of meteorological evaluation.

The authors also thank the reviewer for suggesting inclusion of transportation-related activity data in a Supplemental Information document. We have included these data in graphical format in the Supplemental document for interested readers.

The reviewer also points out areas where our paper seemingly does not align with recent work by Parker et al (2022). Parker et al (2022) is cited in the draft manuscript reviewed by this reviewer, however, the authors acknowledge that we can do more to further compare/contrast the approaches used in these two papers. This will be included in the Discussion section. The differences in results are believed due to subtle differences in spatial scale: our results are presented as basin-averages over the whole SoCAB, while the final sentence in Parker's abstract (pointed out by the reviewer) explores results on sub-basin scales. Figure 10 in Parker et al shows their map of base case O3 minus COVID-adjusted O3 (both modeleed). Spatially, the majority of the study area in their figure shows a decrease in O3, and a small region in LA county shows an increase in O3. On average over the SoCAB, our results appear to be in good agreement. We will make additions to the Discussion section to address these points. The reviewer also brings up Parker's conclusion that parts of the SoCAB were still VOC-sensitive. It should be noted that Parker's basis for this conclusion is outcome-based (i.e. did O3 drop in response to NOx reductions?) rather than process-based (i.e. did the indicators for O3 sensitivity change?). It should be noted that, at the chemical process level, there are numerous scenarios where O3 chemistry may "flip" from VOC-sensitive to NOx-sensitive while still producing an increase in O3 due to non-linearities in chemistry alone, (especially when dealing with airmasses that are near the chemical transition point!) Therefore, Parker's observation that O3 increased in some areas while NOx emissions dropped is not a solid indicator of the underlying chemical regime (*especially* given that the SoCAB is near the chemical transition point!). Our paper presents observation-based evidence that the underlying chemical regime indeed flipped - though we do note in our Discussion section that this may not yield even results over the entire air basin, and that while the basin as a whole is expected to see O3 improvements as NOx is decreased, select areas may see O3 increases in the coming years.

We hope that these additions are satisfactory for the reviewer! Thank you again for taking the time to review our paper.